# Narrowband Theta Investigations for Detecting Cognitive Mental Load

**DOI:** 10.3390/s25133902

**Published:** 2025-06-23

**Authors:** Silviu Ionita, Daniela Andreea Coman

**Affiliations:** 1Department of Electronics, Computers and Electrical Engineering, National University of Science and Technology POLITEHNICA Bucharest, 110040 Pitesti, Arges, Romania; 2Ecological College ‘Prof. Univ. Dr. Alexandru Ionescu’, Intrarea Teilor, No 4, 110029 Pitesti, Arges, Romania; daniela.coman2204@upb.ro

**Keywords:** EEG signal processing, signal metrics, mental state discrimination

## Abstract

The way in which EEG signals reflect mental tasks that vary in duration and intensity is a key topic in the investigation of neural processes concerning neuroscience in general and BCI technologies in particular. More recent research has reinforced historical studies that highlighted theta band activity in relation to cognitive performance. In our study, we propose a comparative analysis of experiments with cognitive load imposed by arithmetic calculations performed mentally. The analysis of EEG signals captured with 64 electrodes is performed on low theta components extracted by narrowband filtering. As main signal discriminators, we introduced an original measure inspired by the integral of the curve of a function—specifically the signal function over the period corresponding to the filter band. Another measure of the signal considered as a discriminator is energy. In this research, it was used just for model comparison. A cognitive load detection algorithm based on these signal metrics was developed and tested on original experimental data. The results present EEG activity during mental tasks and show the behavioral pattern across 64 channels. The most precise and specific EEG channels for discriminating cognitive tasks induced by arithmetic tests are also identified.

## 1. Introduction

Cognitive functions are based on complex brain activity in multiple processes such as sensation and perception, learning, thinking, reasoning, and problem solving, but also imagination and creativity. Thus, a multitude of recognized responsible neuronal areas work cooperatively conjugately or alternatively, in parallel and serially, and this is how we see the effect of the brain’s electrical activity at standard measurement points. Using scalp EEG mapping we can investigate the dynamics of electrical activity in real time in a non-invasive and cost-effective way. The choice of frequency domain for EEG signal analysis is based on evidence from neuroscience as well as practical signal processing considerations. Thus, the theta band (4–7 Hz) is particularly interesting because it involves the existence of several functionally independent activities related to arousal, motivation, cognitive flexibility, and memory-related processes [1]. For the investigation of cognitive mental tasks that involve a state of consciousness, attention, and concentration, the interest in theta waves is precisely related to their specificity in the opposite states, i.e., deep relaxation and sleep [2], as well as in the study of attention deficit [1]. Thus, in our study we chose the theta band considering that it is potentially more precise and specific to cognitive states involving attention, concentration, and working with memory.

The interest in narrowband theta investigations is also motivated by recent studies, which report, for example, that 4 Hz oscillations signify increased motivation (which induces attention), which is reflected by the intensification of the exchange of information between the ACC (anterior cingulate cortex) and the VTA (ventral tegmental area). An increase in the 4 Hz signal has been demonstrated during anticipatory decision making and feedback processing and in the manifestation of behavioral flexibility. Local field potential measurements have shown that in the ACC and VTA, there are peaks of spectral power between 3 and 5 Hz [1]. Additionally, another argument in favor of narrowband investigations is of a practical nature regarding the increased specificity of the analysis of a phenomenon observed strictly on a certain frequency. It will also avoid frequency analysis with the classic FFT technique, applying narrowband filtering instead.

However, the problem of detecting a certain type of cognitive task consists in discriminating against a specific mental state based on the behavior of EEG signals. Thus, the brain’s response to a specific imposed cognitive task must be discriminated against the basic electrical activity related to other nonspecific processes. But this is not simple, for the following reasons: (1) the discontinuity and relative latency of the brain’s response through electrical signals; (2) individual physiological peculiarities that diversify the behavioral pattern of brain waves from individual to individual, depending on several factors (age, gender, education, health status, etc.); (3) the insufficiency of models and detection criteria used in order to classify mental states with very high accuracy; (4) the presence of artifacts in EEG signals; and finally (5) the problem of initial calibration to determine a reference state for mental task detection. Therefore, the identification and recognition of the occurrence of the specific cognitive task is the main task in our research. For this purpose, we proposed a type of mental task of the arithmetic type presented visually, which the subject must solve in the mind, excluding other voluntary motor activities, such as writing and speaking. The actual work task consists of mental numerical calculation operations that are applied continuously, with breaks, respectively, within a series of tests. During arithmetic tasks, a suite of neural processes related to the perception of numbers, the manipulation of numerical quantities, and the decision on the result are carried out, processes that involve memory, attention, and algorithmic and logical processing [3]. The purpose of this paper is to present original results, as a contribution to the methods of analyzing EEG signals for the detection of brain activity during mental arithmetic tasks. The objective of this research is to aim towards BCI technologies for the automatic detection of cognitive brain activity.

## 2. Methodology

The dynamics of the electrical activity of the brain subjected to cognitive tasks involve multiple neural areas and complex signal behavior. As a general goal, we want to detect the brain state during cognitive tasks with non-invasive techniques incorporated into BCI interfaces that are as simple as possible. The working methodology for discriminating certain mental tasks in relation to the background (baseline) activity of the brain includes the following steps: (1) performing experiments with the application of specific tests to induce the specific cognitive mental state; (2) acquiring and exploring EEG signals at a sufficient number of points on the surface of the head; (3) establishing relevant signal measures (metrics) for real-time analysis to detect the cognitive task; (4) choosing task discrimination/recognition criteria in relation to a reference mental state; (5) calibrating the system to establish the reference state.

This paper presents a new way of investigating EEG signals and develops an algorithm that can distinguish as precisely as possible the brain activity specific to arithmetic calculation, without the use of machine learning and deep learning methods. Our approach is based on calibrating the system correctly during the experiment on the test subject. Thus, the uniqueness of any subject is considered, and this allows the application of the method to different subjects by individual recalibration based on the EEG acquisition even in the current state preceding the task. Unlike approaches based on machine learning, this algorithm applies an analytical approach to analyzing signals in real time, without requiring large volumes of data for training. In contrast to the shortcomings of machine learning-based techniques, the proposed method has the following advantages: 

- It does not require large amounts of preprocessed, labeled data; 

- It does not depend on models trained on certain groups of subjects, performed under laboratory conditions; 

- It is based on a transparent, interpretable detection model and can therefore be controlled through its parameters that are explicit; 

- It provides better real-time responses with reduced computing resources; 

- It provides greater confidence in conditions of noise, motion, or other biological and non-biological artifacts.

Inter-subject variability is a critical factor influencing the performance of mental load detection models and especially limits machine learning-based recognition algorithms. This obstacle led us to approach an analytical model detection method instead and to limit ourselves to investigating a single subject through multiple tests.

### 2.1. Test Design

For experimentation, we proposed a number of cognitive stimulation tests consisting of mental arithmetic tasks. The tests consisted of arithmetic calculation exercises, displayed on the screen with a predetermined time limit. In order to maintain the subject’s attention and concentration on the main task, the calculation operations consisted of successive and recursive subtractions and additions, starting from different numbers, with a subtraction/addition rate of 2, 3, 4, 5, or 6 units (for example: 100 − 2 − 2 − …, 60 + 3 + 3 + …, 32 + 4 + 4 + …, 41 + 5 + 5 + …, or 41 + 6 + 6 + …). The datasets collected correspond to the following types of tests:–6 tests without task, with eyes open (looking at a white display), called TW1 to TW6.–6 no-load, eyes-closed, relaxed tests, called T01 to T06.–6 continuous arithmetic task tests, called TC2, to TC7.–6 alternating arithmetic task tests, designated TI5 to TI10.

Each task test included a 10 s break at the beginning and a break of at least 5 s at the end. The intermittent task tests were organized with three calculation rounds with 10 s breaks between them, while the continuous task tests had five successive calculation rounds without a break between them, as illustrated in Figure 1.

### 2.2. EEG Exploration

Exploration of the electrical activity of the brain in the experiments was carried out by acquiring EEG signals using the Biosemi Active Two system, configured with 64 channels in monopolar mounting, positioned and labeled according to the international standard “10-20” [Biosemi EEG ECG EMG BSPM NEURO amplifier electrodes]. The equipment used provides a large amount of data from the 64 measurement points that provide good spatial resolution. The EEG signals are sampled at a frequency of 8192 Hz, which allows the acquisition of neuronal impulses on the scalp surface with good temporal resolution. The raw original signals were acquired in the frequency band set to 0.16 Hz to 40 Hz, digitally converted to 24 bits and stored in files in the bdf (Binary Data File) format. An experimental instance and the map of the measurement points are shown in Figure 2.

The experiment involved an adult subject, 58 years old, male, with professional activity in the field of engineering. The subject participated in 18 consecutive sessions, each lasting approximately 1.5 minutes, totaling approximately one and a half hours of participation, including technical breaks. 

As an additional measure of protection against possible electromagnetic disturbances during the acquisition, the subject was placed together with the equipment in a metallic shielding enclosure. No device powered by a switching power supply was located inside, as all the devices involved there were powered by batteries. Data transmission to the external process computer as well as acquisition control were performed via optical fiber, through the specific interface from Biosemi. The experimental procedure included the preliminary stage of subject preparation and the application of the measurement electrodes. The subject was previously instructed on the structure of the tests and the type of arithmetic tasks to be performed. The tests described above were visually displayed with PowerPoint, triggered by the subject by pressing a key—the only motor action performed by him. After one second, a short sound signal was emitted and the external operator commanded the start of the acquisition on the process computer. At the end of the test, a specific acoustic signal would announce the stop of the recording. There was no automatic triggering and stopping of the tests and thus the duration of the recordings may have differed slightly from one test to another, but this did not affect the integrity and structure of the useful data. Usually there was a small variable latency when stopping the recording, which slightly affected the final quiet period. During the tests, the subject complied with the preliminary instructions: to be as relaxed as possible, not to move, to blink as rarely as possible, to keep his gaze fixed on the center of the screen, and to control his thoughts and mental state in accordance with the task—that is, to be absolutely focused on the calculations when they appear, respectively, and to relax mentally as much as possible during the break periods. The relaxation and break periods are spent with the eyes open, without natural ambient light, except from the white background displayed by the PowerPoint application during the test. We highlight the fact that in these tests, the relaxation periods, breaks, and periods of mental load are part of the same recording, which limits the effects of subject variability. This supports the proposed detection method by bringing the relaxation period used as a reference for calibration as close as possible to the subject’s current mental state and physiological condition. In contrast, machine learning-based reference models are not actually personalized and are susceptible to failure.

## 3. Principle of the Method

In this section we describe the last three points of the proposed working method for signal processing and analysis, which concern the following: signal metrics, mental task discrimination criteria, and the calibration method for the reference state Section 3.1.

### 3.1. Signal Metrics

The considered metrics usually determine the signal analysis method. In EEG, parameters such as signal energy, number of zero crossings, or other transient signal behaviors are used in the time domain analysis. Parameters related to signal frequency, such as spectral power and its distribution, which are typically based on Fourier analysis and transformation, are used in the frequency domain analysis. Also, combined time–frequency methods provide specific distributions that are useful in EEG interpretation. Finally, the probabilistic behavior of signals can be quantified with known statistical parameters within the framework of statistical analysis methods. These analysis techniques, which are widely used in EEG studies, constitute the majority of reports in the literature dedicated to the field. We mention here some of those that address the issue of recognizing electrical activity of the brain in the presence of mental stress induced by mathematical tasks. Thus, in references [4,5] the use of signal energy and statistical parameters as metrics to detect arithmetic mental load is reported, and the analysis in frequency and time–frequency domains is addressed in [6,7]. The use of statistical measures together with machine learning techniques is reported in [8,9], while several works address spectral analysis based on FFT, the spectral power distribution indicator, including in combination with statistical indicators [10,11,12,13,14]. We also mention here a previous work by the authors of this research [15], in which we reported some results obtained with the statistical correlation method and the spectral analysis method based on FFT evaluation. On the other hand, alternative brain scanning methods that do not use EEG signals, such as functional magnetic resonance imaging (fMRI) and functional near-infrared spectroscopy (fNIRS), use specific image analysis techniques [16]. In our study we introduce as a measure for the analysis a signal parameter consisting of the length of the signal curve over a period. Since the period of EEG signals is variable in the dedicated analysis bands, we build our metric for a narrowband signal that we obtain with a suitable bandpass filter (BPF), with the median frequency $F_{{med}_{BPF}}$. Thus, the filtered signal for analysis will have a constant period $T_{med}=1/F_{{med}_{BPF}}$ and by sampling with the frequency $F_{s}$, one period will contain $n=T_{med}\cdot F_{s}$ signal discrete points. For the discretized signal $S\left(t_{k}\right),\;k=1,\ldots n$ with the sampling frequency $F_{s}$, we will accept for the beginning the expression for calculating the length of the signal curve over the period *p*, as follows:(1)$$Lsc\left(p\right)=\sum_{k=1}^{n}\sqrt{{{(∆}_{k}S)}^{2}+{\left(\frac{1}{F_{s}}\right)}^{2}}$$

Even though the relation is not dimensionally homogeneous, it expresses a sum of elementary Euclidean distances and therefore the path traveled by the signal in the signal–time plane during a period. Since $∆t=1/F_{s}$ is the (constant) time step of discretization, the relation can be reduced to the following form:(2)$$Lsc\left(p\right)=∆t\sum_{k=1}^{n}\sqrt{{\left(\frac{{∆}_{k}S}{∆t}\right)}^{2}+1}$$

*Lsc* expresses the length of a path parameterized in R2, where the parameter is the time variable itself, which can be evaluated with the discrete formula of the first-order curvilinear integral. However, we would prefer to normalize this parameter by referring to $T_{med}$, successively obtaining the following relations:(3)$$Lscn\left(p\right)=\frac{∆t}{T_{med}}\sum_{k=1}^{n}\sqrt{{\left(\frac{{∆}_{k}S}{∆t}\right)}^{2}+1}=\frac{F_{med_{BPF}}}{F_{s}}\sum_{k=1}^{n}\sqrt{{\left(\frac{{∆}_{k}S}{∆t}\right)}^{2}+1}=\frac{1}{n}\sum_{k=1}^{n}\sqrt{{\left(\frac{{∆}_{k}S}{∆t}\right)}^{2}+1}$$

In this form, the parameter length of the signal curve during a period, is equal to 1 at the limit, which represents the minimum value that is reached for the constant signal. The parameter *Lsc* depends on quantitative characteristics of the signal—amplitude and dynamics but is also sensitive to the shape of the signal affected by transient local variations. However, it is not influenced by the level (bias) of the signal, which gives it immunity to the non-stationary behavior of signals. It is a good indicator of the signal energy provided by the signal dynamics and not by its level. Finally, it is specific only for narrowband filtered signals and thus for a nominal frequency analyzed. Other practical benefits of the proposed signal parameter are related to the relatively reduced processing effort, which no longer involves the calculation of the Fourier transform for laborious spectral analyses in wider frequency bands. The signal energy for a period with n samples is a measure with the absolute value given by the following equation:(4)$$Ea\left(p\right)=\sum_{k=1}^{n}{\left(S\left(t_{k}\right)\right)}^{2}\;$$

Unlike *Lsc*, the energy also depends on the signal bias in the period *p* and will be considered as an additional criterion for discriminating mental tasks strictly in the narrow working band. However, the current signal level may also depend on other causes not specific to the mental task under study. Therefore, we will use the relative value of the energy calculated in relation to the average signal over the current period *p*, denoted by ${\left(\bar{S\left(t_{k}\right)}\right)}_{p}$, according to the following relationship:(5)$$Er\left(p\right)=\sum_{k=1}^{n}{\left(S\left(t_{k}\right)-{\left(\bar{S\left(t_{k}\right)}\right)}_{p}\right)}^{2}\;$$

### 3.2. Detection Criteria and Conditions

Building mental load detection criteria takes into account the signal measures discussed above for the calculation of detection thresholds and the formation of discrimination/recognition conditions of the mental state for each measured EEG channel *i =* 1,…64. The calculation of detection thresholds is the subject of the system calibration operation, preceding the acquisition of signals in the cognitive load experiments. The basic idea of the mental load detection algorithm is the dynamic processing of the acquired data and the comparison of the discrimination characteristics with detection thresholds calculated on a portion of the beginning of the signals, within a time interval allocated for calibration. This solution does not use machine learning techniques that require large volumes of training data and the performance of many EEG signal acquisition experiments. Unlike the pattern recognition technique which functions by referring to a model learned from multiple examples, the proposed method is limited to the preliminary processing of a signal portion with duration $t_{calib}$ from the beginning of each EEG signal captured from the subject in a voluntary state of mental rest (relaxation). Therefore, the detection thresholds are defined based on the averaging of the signal characteristic (*Lscn* or *Er*) in the window with a set number of periods *p =* 1,…,*w*, which moves incrementally by one period throughout the calibration duration *t_calib_*, finally selecting the maximum value found, as follows:(6)$$Tresh1\left(i\right)=\underset{t_{\mathrm{calib}}}{\max}\left({\left(\bar{Lscn\left(p\right)}\right)}_{p=1,w}\right),$$
and(7)$$\mathrm{Tresh}2\left(i\right)=\underset{t_{\mathrm{calib}}}{\max}\left(({\bar{Er\left(p\right)})}_{p=1,w}\right),$$
where w is the number of signal periods that defines the width of the averaging window for the signal characteristic being evaluated. The use of detection thresholds is conducted with the following relations:(8)$$\left({\left(\bar{Lscn\left(p\right)}\right)}_{p=1,w}\right)>Tresh1\left(i\right),$$(9)$$\left({\left(\bar{Er\left(p\right)}\right)}_{p=1,w}\right)>Tresh2\left(i\right),$$
applied iteratively with a step of one period to calculate the current values of the averaged signal measures if the current time $t>t_{calib}$. Thus, after the calibration period, following the processing of the signals on each EEG channel i in successive windows with w periods that move incrementally with a step of one period, current values of the averaged signal measures are obtained, which are compared with the threshold values determined during calibration according to relations (8) and (9). The classification of the current mental state associated with the presence or absence of mental load is therefore made based on the threshold limit conditions. If one of the above conditions is true, then it is considered that the mental load has been detected according to the respective criteria, and the presence of this state will be signaled.

### 3.3. Detection Algorithm

The detection of cognitive mental load in the low theta band begins with the processing of EEG signals by filtering. For this purpose, two filters were designed: the first, a lowpass filter (LPF) with a cutoff frequency of 2 Hz to extract slow components from the raw EEG signals, and the second, a bandpass filter (BPF) for the narrow band (4–5 Hz) of the low theta wave domain. These are high-order FIR digital filters. The first filter has the role of providing a signal with a slow variation below 2 Hz that will be subtracted from the raw signal to reduce the effect of level drift. The second filter provides the signal component in the narrow frequency band of interest for analysis.

Artifact removal is a generally necessary step for analysis in wider frequency bands. In the case of narrowband analysis, even after eliminating the effects of low-frequency components, the problem arises of eliminating spike-type disturbances that are generally more visible in raw data when the sampling frequency is high. Usually, resampling the signal with half the initial frequency reduces these signal spikes acceptably. In our case, the signals are sampled with $F_{s}=8192\;Hz$, which is of high value, and we do not want to lose this advantage. Therefore, we proposed an algorithm for treating spike-type disturbances using a condition for their detection and a rule of replacement and not of elimination itself. The working principle of the designed algorithm is stated as follows: if the signal peak in the current analysis period exceeds a threshold set in relation to the average of the signal peaks in the previous signal window, then the current signal period is considered a disturbance and is eliminated by replacement. Elimination by replacement ensures a local signal uniformity as an effect of replacing the current signal period, as well as a number of neighboring periods, with the average of the signals in the previous window. The working steps of the proposed algorithm for treating peak disturbances are structured as follows:

{1} The artifact detection condition is defined by the signal peak criterion (SPC):(10)$$SPC=\frac{Speak\left(p\right)-({\bar{Speak\left(p\right)})}_{p=p-np,\ldots ,p-1}}{({\bar{Speak\left(p\right)})}_{p=p-np,\ldots ,p-1}}>Tresh,$$
where *np* is the number of periods considered in the window preceding the current signal period *p*, and *Tresh* is the value set for artifact detection.(11)$${\left(\bar{Ssp\left(p\right)}\right)}_{p=p-np,\ldots ,p-1}$$

{2} If condition (10) is true, the signal segment corresponding to period p, Ssp(p), respectively, is considered an artifact and is eliminated. The elimination rule is applied as follows: replace the artifact signal segment Ssp(p) as well as the neighboring segments Ssp(p − 1), Ssp(p + 1), Ssp(p + 2) with the average value of the signal in the window preceding the artifact.

Now, the entire algorithm for cognitive load detection can be run according to the following steps:

(1)Starting the acquisition of raw EEG data on 64 channels (in the 20-10 system).(2)Starting data processing after a stabilization time interval (t_start_ = 0…2 s).(3)Data normalization (equivalent to signal conditioning).(4)Applying the correction filter to reduce the slow variation of the signals (below 2 Hz);(5)Narrowband filtering (BPF application) and data retention after 1 s.(6)Calculation of the average period of the filtered signal based on the average frequency of the BPF passband: $T_{med}=1/F_{med_{BPF}}$.(7)Calculation of the number of signal samples from an average period: $n=T_{med}\cdot F_{s}$, where $F_{s}$ is the sampling frequency of the EEG signals (8192 Hz).(8)The current signal is dynamically segmented into frames equal to the average period—containing *n* samples: $Ssp\left(p\right)={\left(S_{k}\right)}_{p}$, for k = 1, …, n and p = 1, …, Np, where Np is the number of periods in whole signal.(9)Application of an algorithm for automatic artifact removal.(10)The signal measures are iteratively calculated for each period *p* equal to *Tmed*, respectively, *Lscn(p)* with relation (3) and *Er(p*) with relation (5).(11)The calculated characteristics are averaged over a sliding window containing w signal periods: ${\bar{Lscn\left(p\right)}}_{p=1,w}$, respectively, ${\bar{Er\left(p\right)}}_{p=1,w}$. The width of the signal window for averaging is chosen as w = 3…20 periods.(12)The detection thresholds are calculated as the maximum value of the averages of the characteristic over the established calibration period $t_{calib}$, for each channel *i* = 1, …, 64 measured, with the relations (6) and (7). The calibration period is at least 10 s, at the beginning of the EEG signal acquisition session; a state of imposed mental rest is mandatory, usually with eyes open.(13)After the calibration period has elapsed, the mental task discrimination conditions are evaluated on successive signal windows with width w, in the form of the criteria expressed by the relational expressions (8) and/or (9).

This algorithm was implemented and tested using MATLAB (version 2023b), including the Signal Processing Toolbox, as well as the additional library FieldTrip [17], for reading BDF format data recorded with the Biosemi equipment. In our study, we used this algorithm in an application program to investigate the already recorded data, working offline with the EEG acquisition equipment. However, the program is designed modularly so that it can be integrated with the specific acquisition equipment SDK (from Biosemi, in our case) and thus exploit the real-time mental load detection algorithm capacity.

We applied the cognitive load detection algorithm on the data recorded, with the following parameters set:

-Narrow analysis bandwidth 4–5 Hz;-Average window size w = 10 periods;-Calibration duration $t_{calib}$ =10 s;-Signal metrics used *Lscn*, and *Er* only for some comparisons.

In the data preprocessing stage, steps (3)–(5) are effectively considered. Thus, in the offline working mode, in step (3), the data were read using the ft_read_header and ft_read_data function pair from the FieldTrip toolbox, after which they were normalized using the z-score method, in the range [−1, 1]. Then, in step (4) the lowpass filter was effectively applied in the baseline correction and also to exclude certain biological and non-biological artifacts. The lowpass filter is of the digital IIR Butterworth type with a pass frequency of 2 Hz and a cutoff frequency of 2.25 Hz. It extracts the slow signal, which is then subtracted from the raw signal. In step (5), narrowband filtering specific for mental load detection is applied. A high-order FIR digital filter, more precisely 4000, with a narrow passband from 4 to 5 Hz was used. This results in the signal to be analyzed for mental load detection following steps (6)–(12). In step (13), the calibration is already completed, and the algorithm continues its execution cyclically, on successive signal windows with the application of the atypical peak artifact elimination model, according to Equations (10) and (11) and the testing of the load conditions with Equations (8) and (9). The answer consists of the number of occurrences of the detected cognitive load, which are counted and displayed graphically for all EEG channels.

## 4. Results and Discussions

To prove the proposed concept for narrowband cognitive task detection, we exploited the open-access Physionet international database Electroencephalograms during Mental Arithmetic Task Performance [https://www.mdpi.com/2306-5729/4/1/14, accessed on 1 March 2025]. This provides a sample of 34 valid recordings selected from an initial experimental sample of 66 young participants aged 16 to 24 years, both male and female, who were subjected to mental arithmetic tests based on serial subtraction. The data are provided from a 19-channel EEG system in the 20-10 system, with the following EEG channels: Fp1, Fp2, F3, F4, F7, F8, T3, T4, C3, C4, T5, T6, P3, P4, O1, O2, Fz, Cz, and Pz. By processing the data, we generated two datasets for each EEG channel: the first from the no-load recordings during a relaxation period with eyes closed (60 s) and the second from the recordings with the computational load applied (the next 60 s), for the entire sample of participants. To evaluate the effect of the proposed task detector, we applied the standard statistical *t*-test. The test hypotheses concern the significance of the difference between the statistical means of the two datasets containing the number of activations (cognitive task detections) on each EEG channel, obtained from each participant, the first set without the cognitive task, and the second with the application of the mental task. Table 1 presents the results of the *t*-test for two confidence levels α = 0.05, respectively, and 0.01 with the validation of the hypothesis H1—“the difference between the means is significant”. The true value for this hypothesis occurs if |t| > the critical value from the t-table, which means that the respective EEG channel could be reliable for the detection of the cognitive task. The critical values in the *t*-test table for the parameter degree of freedom df = 33 were calculated by linear interpolation between the values available in the *t*-test table that correspond to the neighboring values df = 30 and df = 40.

Therefore, the EEG channels sensitive to the detection of cognitive computational load are Fp1, Fp2, F3, F4, C4, and P3 with the 95% confidence level, and Fp1, F3, C4, and P3 with the 99% confidence level, respectively. For a lower level of accepted confidence, for example 80%, obviously more EEG signals become potentially eligible for the detection of cognitive load, as follows: Fp1, Fp2, F3, F4, F8, T3, C4, T6, P3, and O2. For all channels found to be reliable, the results are statistically significant, with the calculated *p*-value being less than 0.02. A summary statistical representation with a box plot can be seen in Figure 3. It is observed that, in general, the eligible channels with a high confidence level do not contain data with outliers, those marked with a circle in Figure 3.

In relation to what was found in this analysis, we mention that the channels found to be sensitive to the cognitive computational task are among those found with other conventional signal metrics, or novel combinations thereof. A synthetic review of the results reported in the literature can be found in the paper [15]. We synthetically highlight the particularities of performing the experiments in the Physionet database [18], as follows:The sampling frequency of the signals is 500 Hz.The 19 EEG signals captured in the 0.5–45 Hz frequency band were recorded and previously cleaned of artifacts.The computational task was communicated to each participant verbally by acoustic means.The participants were in the reference state (relaxation without cognitive load) with eyes closed.The computational task was processed continuously without voluntary interruptions.

Our own experiment comes with particularities that complete and add more information, as follows:-The signal sampling frequency is 8192 Hz.-The number of electrodes used for measurements is 64 in the 10-10 system, which provides a higher EEG mapping resolution; at least, this is expected.-The task detection algorithm is designed to work on-line with the subject, in real time as much as possible; so we avoided using an artifact removal technique that requires intensive computing resources (ICA for example). We applied our own simpler algorithm (see in Section 3.3), which was close to the wavelet-based principle. In addition, we applied a rejection of slow components below 2 Hz by extracting them with the lowpass filter, followed by subtracting these components from the raw signal. Moreover, the narrowband working principle of the proposed method has the advantage of eliminating some biological and non-biological artifacts.-During the experiments, the computational task is exclusively communicated to the subject visually by displaying it on the screen.-The experimental sample includes six repeated measurements of a single participant under three test conditions: (i) without imposed cognitive load, (ii) with continuously applied cognitive load, and (iii) with intermittently applied cognitive load, with precisely determined breaks. We argue this approach as follows:
(a)A multitude of variables that differentiate subjects in an experiment are eliminated, in relation to the following: age, gender, level of education, cognitive abilities, emotional control, attitude towards the experiment itself and the degree of involvement, some physiological, psychological, and general health characteristics, etc.(b)A single subject trained and appropriately motivated has better controlled behavior during the experiments, which gives a higher degree of confidence in the results.

In the experiment, the tests were classified and named as follows: 

- Tests for the reference state, without cognitive load TW1 to TW6, 

- Tests with continuously applied cognitive load TC2 to TC7,

- Tests with intermittently applied cognitive load TI5to TI10.

The *t*-test works quite well when the sample size is small, so in our case we applied it twice on paired data samples: the first time for continuous load versus no load, and the second time for intermittent load versus no load. For both cases, the EEG activity on all 64 channels is statistically represented in Figure 4a,b.

At a glance we see a fairly correct statistical distribution, with few outliers and median values significantly detached in several EEG channels. We applied the *t*-test for three confidence levels α: 0.05, 0.02, and 0.01. Table 2 presents the results that verify the alternative hypothesis H1 for the continuous cognitive task. Table 3 presents the results for the intermittent load tests.

The results show that for both modes of load application (continuous or intermittent), even at a high confidence level of 99%, the detector was able to discriminate cognitive activity induced by the mental calculation task from background brain activity, without the task, with eyes open. This is possible for a few EEG measurement channels located over recognized portions of the cortex with responsibilities in numerical calculation skills. It is evident that the significant EEG channels are increasingly numerous, as the accepted confidence level decreases. We note that from the point of view of statistical significance, some channels do not meet the *p* < 0.05 condition, such as FT7, FC3, C1, C5, CP5, and PO7 in Table 2 and FC1, F2, FC4, P10, and PO7 in Table 3. At this stage, they are not considered. Table 4 shows the significant channels that are successively added from bottom to top to the initial set found for the highest confidence level.

The EEG channels that stand out from the background in the presence of cognitive load cover the prefrontal region that governs executive operations for mental control responsible for working memory and attention. Both processes are involved in arithmetic calculation tasks in specific tests with serial subtraction or addition operations that are equivalent to counting backward or forward by an arbitrary step, starting from a specified number.

The methodology used is the comparative analysis of the results of the two-step test process quantifying the following indicators:(i)The activity of each EEG channel in the absence of the load, by the number of validations of the detection condition with the calibration reference threshold calculated in the first 10 s.(ii)The activity of each EEG channel in the presence of the cognitive load, by the number of validations of the detection condition with the calibration reference threshold calculated in the first 10 s.

The first indicator, from step (i), gives us information about the activity of the EEG channels caused by causes other than the imposed cognitive load. In a sense, it shows us how stable the imposed (relaxed) state is in non-load tests and gives us information about the background brain activity on each EEG channel. Basically, it is necessary to discriminate the cognitive load state from the background state (generated by other causes).

The second indicator, from step (ii), provides the quantitative response on each channel in the presence of the task, but without excluding false positive responses generated by other causes.

In Figure 5 we provide as an example the detected activity of each channel for the reference no-task test with eyes-open T1. In Figure 6 the evolution of the cumulative activity of the EEG channels during the test is shown, which reflects the fluctuation of the stability of the (relaxation) state imposed during the test. It is observed that most of the EEG channels are active in all areas of the head and, cumulatively, they manifest themselves throughout the test, which proves that the specific relaxation state imposed is not exactly stable in the sense of uniformity. This fact can be attributed to background neural processes, as well as to involuntary mental states. From the point of view of the specific cognitive mental task, these represent responses in the false positive category. The most active and fluctuating signals in the relaxed states were detected in the parietal and occipital areas, in both cerebral hemispheres. The least active channels are detected in the frontal and prefrontal areas and only a few in the central and parietal and occipital regions.

In the case of the tests without load with eyes closed, the activity on the channels has a different appearance than that in the test with eyes open. The study of these experiments will be addressed separately in a future paper. Next, we approach the second step for evaluating the activity on the EEG channels in the presence of the imposed cognitive load. For exemplification, the quantitative graphs with the number of activations on each channel cumulated during the tests with the load are shown in Figure 7 for a few tests with continuous load and in Figure 8 for some tests with an intermittent load.

The main objective is to identify active EEG channels that, on the one hand, are specific and sensitive to the imposed cognitive load and, on the other hand, reflect with maximum precision the presence of cognitive load. This problem has been addressed in the literature with the aim of optimizing the number of electrodes used for signal capture [6,13]. A first level of validation of the relevant detection channels was achieved with the *t*-test, which confirmed the specificity of some channels. Now we bring the issue of real-time detection; in this case we need to analyze the accuracy and sensitivity of detection. Here are the dynamics of the application of the cognitive task matters, to see the response of the detector in relation to the task profile during the test.

The confidence in the algorithm’s ability to discriminate mental load is assessed with the set of known criteria, defined as follows:(12)$$Accuracy=\frac{TP+TN}{TP+FP+TN+FN}$$(13)$$Precision=\frac{TP}{TP+FP}$$(14)$$Sensitivity=\frac{TP}{TP+FN}$$(15)$$\mathrm{Specificity}=\frac{TN}{TN+FP}$$

If we merge relations (14) and (15) with (12), we obtain the accuracy in the following suggestive form:(16)$$Accuracy=\frac{TP+TN}{\frac{TP}{Senzitivity}+\frac{TN}{Specificity}}$$

We mention that these indicators are based on the probabilistic quantities that quantify the detector response: true positive (TP), false positive (FP), true negative (TN), and true positive (TP). In principle, they depend on the number and timing of response occurrences due to the fulfillment of conditional Equations (9) and (10). The ideal situation of maximum detection accuracy is when sensitivity and specificity simultaneously reach the maximum value of 1, and implicitly the precision will also be maximum. It is understood that the key factors are the rates of FP and FN. We applied a procedure to select EEG channels that meet the accuracy and precision parameters in discriminating the cognitive task to the greatest extent possible with the following steps:

(1) The channels classified as significant with the *t*-test are considered. These are those in Table 4. Statistically, they have been shown to be the most specific for the detection of cognitive load and provide a high degree of confidence. However, false positive responses are still possible, and this influences the detection precision.

(2) The confidence criteria are calculated with Equations (12)–(15), and precision is given priority. The highest values are eligible, the maximum being 1. In this stage, those channels with FP = 0 responses are identified, that is, those that have maximum precision and specificity. However, this situation is an ideal one; so the decision to choose a certain acceptable level of precision as well as sensitivity bears some discussion in the following.

The evaluation of false positive responses is conducted in relation to the temporal application profile of the cognitive task within the tests. Figure 9 shows the activity on channels 1 to 5 in the TI7 test with the intermittent task, where a general synchronization of the detector responses (red bars) with respect to the periods of application of the cognitive task (green lines) is observed. We recall that the statistical test validated channels 1, 2, and 5 at the 95% confidence level.

The responses strictly overlapping with the task lines are the precise and specific ones. Other responses outside the task periods are also observed that are classified as false positive, and this disqualifies the channels from being precise and specific. Indeed, part of the false positive responses may be due to causes that could not be excluded by the reference test or during the calibration period, but some of them may also be due to the dynamics of the neural processes specific to the cognitive mental task. Thus, prolonged responses beyond the timing of the cognitive task (see channels 1, 2, 3, and 5) can be attributed to natural latencies of mental processing. Even other transient responses that occur recurrently during the pause periods between tasks could be attributed to cognitive aftereffects or nonspecific involuntary states (see channels 1 and 2). Under these circumstances, we could consider acceptable channels that show prolonged responses or recurrences within a certain time interval after the cognitive task has disappeared. We will call this time interval the *accepted latency*, and we will consider for it a cutoff value of 3 s. In this way, for example, channels 3 and 5 become accepted as precise and specific.

Another problem of detection is the continuity of responses, which is related to the sensitivity of the EEG channel to the specific cognitive task and which, implicitly, affects the accuracy of the detector. The issue of the sensitivity of the EEG channel is related to the number of true positive (TP) activations in relation to the imposed cognitive load; thus, the greater the number of TP responses or, conversely, the smaller the number of FN responses, the more sensitive the channel is. Figure 9 shows that channels 1 and 2 are more sensitive than the others and that channel 4 is the worst.

A good detector should have maximum precision and specificity, as high a sensitivity as possible, and this would also guarantee as high an accuracy as possible. In Figure 10 we have provided the result of the evaluation of the indicators discussed above, for three tests from both types of cognitive load.

Table 5 shows the numerical ranges for the performance criteria calculated for those specific EEG channels selected with the *t*-test for the highest confidence level (99%), for intermittent cognitive load applied.

These deterministic values are based on probabilistic values regarding the occurrence of the task detector response with respect to the task profile, which is strictly defined in time. But the neural dynamics are quite complex and there is no guarantee that the EEG signals should perfectly copy the cognitive task profile. Indeed, statistical tests show good detection specificity for certain signals, and this can even be seen visually in Figure 11, Figure 12, Figure 13 and Figure 14.

Gathering the information revealed so far, we can make the following interpretations:

The EEG channels with high specificity (a confidence level of 98%) are located in cortical areas recognized for mental processes in the presence of cognitive tasks in general, and numerical ones in particular. Thus, in the frontal lobe, when the intermittent task is applied, the channels AF7, F5, F7, T7, and TP7 become active in the left hemisphere, and in the right hemisphere, Fpz, Fp2, AF8, AF4, F4, F6, FT8, T8, and FC6 become active. During tests with a continuously applied cognitive task, AF7, F5, and F3 were active in the left hemisphere, and TP8 and P10 were additionally activated in the right hemisphere. In the tests based on Physionet, the task was applied continuously, but as the number of measurement points was smaller, channels located in the same frontal areas were detected: Fp1 and F3 on the left, and Fp2 and F4 on the right, but in addition, the P3 channel was also detected in the left parietal, and C4 in the right central area.

This is remarkable in all tests to detect EEG activity in the frontal areas that govern executive operations for the mental control responsible for working memory and attention. Both processes are involved in arithmetic calculation tasks in specific tests with serial subtraction or addition operations that are equivalent to counting backwards or forwards by a certain step, starting from a specified number. The signals coming from the temporal and frontotemporal areas (T7, T8, FT8, TP7, and TP8) signify work with declarative memory. These do not manifest themselves in the same way in all tests, probably due to the variable mental effort. An interesting signal, detected only in the Physionet experiment, is P3 in the parietal region, where the existence of a so-called hub of computational abilities was recognized [19]. In our experiment this channel was not eligible according to statistical tests; in some tests it appears with low sensitivity and therefore has little activity. However, in tests with continuous load the activity of the opposite channel (in the right hemisphere) P4 is noted, but only for the 95% confidence level.

A remarkable fact is related to the differences in EEG activity in tests with intermittent load versus those with continuous load, where a certain balance appears in the left side of the frontotemporal area (including F7, T7, and TP7) in the case of intermittent load.

A more accurate decision on the choice of the most suitable EEG channels for a specific task can be made based on a larger experimental study. In any case, with the procedure described above, our study reveals distinct patterns of EEG activity on the selected channels, in correlation with the profile of the applied cognitive task. Thus, in Figure 11, we can clearly distinguish in a large part of the channels the gaps in EEG activity synchronized with the intermittent application of the mental task (tests T5 and T6). The T7 test is presented in Figure 12, the left stack, and in the right stack we can see the comparative EEG activity in the absence of the cognitive task in the T1 test. Also, in Figure 13 we see the EEG activity without regular gaps when the task is applied continuously in tests T2 and T3, respectively, in Figure 14 for the T4 test. In addition, in tests with continuous tasks we find an increase in the activation rate over time, as the mental task progresses, which suggests the effect of progressive brain loading.

These results are encouraging for our stated goal in the introduction of the paper, i.e., to detect cognitive load associated with arithmetic calculation without involving machine learning techniques and large datasets. The presented experiments proved that the detection of mental load based on an individual threshold—as a reference state, using a distinct signal metric and analyzing the signals in a specific narrow band—is feasible. The discussion of the results involves several panels.

First, the operation of the method was proven on the Physionet database and then on the datasets acquired through our own experiments. In both cases, the method proved capable of discriminating mental load in relation to the reference resting state despite the differences in equipment and test format. Regarding the performance of the load detector, we found that the duration of the calibration period positively influences the detection accuracy, but a duration that is too long may include disturbances in the individual’s resting state, which was found in some recordings from Physionet where the last 40 s of rest before applying the mental load was taken for calibration. In our experiments, the subject undergoing tests was better controlled, in the sense of respecting the state imposed by the experimental protocol. In this case, this is an advantage of performing the tests on a single individual. The duration of the reference state imposed for calibration was only 10 s. Another parameter that influences the accuracy of the method is the size of the signal window in which the metric used for detection is averaged. The window comprises a specified number of signal periods over which the averaging is performed, and the size of this number is the subject of a compromise: a short window benefits real-time detection but degrades detection accuracy, and, conversely, a longer window increases accuracy but introduces a corresponding delay. We obtained the presented results using a signal frame containing 10 periods, which means about 2.2 s of response delay and the best detection accuracy of around 0.7 (as seen in Table 5). Increasing the frame length to 20 periods results in an inherent delay of over 4 s, but also an increase in accuracy to about 0.85.

Secondly, the relevance of the results from an experimental point of view is supported by the evident synchronization of the activation of some signals with the profile of the application of the cognitive task over time. Thus, our experiments were structured in such a way as to stimulate specific mental activity dynamically and intermittently in the narrow band chosen for analysis.

Third, the sustainability of the results from a neuroscience perspective comes from the statistical analysis of the experimental data which revealed a high level of confidence for certain signals with a confirmed role in the detection of mental load, as presented in Table 4.

## 5. Conclusions

In this paper, we presented the results of an original experimental study highlighting the EEG activity of the brain during the processing of a cognitive arithmetic task. This investigation is a complementary contribution to our previous work in the field, the results of which have already been reported [15]. In this study, we used our own mental task tests. The research methodology had two objectives: first, to create an EEG signal analysis algorithm for the detection of the specific cognitive task in real time, and second, to identify the most reliable EEG channels for the applied cognitive task. The study avoids the use of machine learning techniques that require large volumes of training data, which must be as diverse as possible—practically requiring the performance of numerous EEG experiments. Thus, instead of using pattern recognition through pre-trained models from multiple examples, the proposed method initially processes a short portion of the EEG signal in a state of imposed relaxation, to perform a calibration within the current monitoring, just before applying the cognitive task. This processing allows the establishment of reference threshold values for each EEG channel, to which the sequences of dynamically captured signals are subsequently related. The proposed signal metric is an original quantity represented by the length of the signal curve in portions equal to the signal period and which is related to the number of samples of the period. This metric is applied to the very narrow band of the filtered signal, with a width of 1 Hz. Thus, the proposed metric characterizes the signals not only from an energetic point of view, but also from the point of view of the shape of its function (curve). It does not give the signal energy itself, which can be evaluated as a separate measure, but depends on it. The use of the relative signal energy as a characteristic measure for task discrimination revealed similar results, with some small differences in the number of activations on certain channels. In summary, the main results of our research can be described as follows:-We demonstrated narrowband EEG activity at low frequency in the theta wave domain for arithmetic cognitive tasks.-We demonstrated the feasibility of an algorithm suitable for real-time detection of a cognitive task, using a particular signal metric—the length of the signal curve over a period, with detection criteria of the reference threshold type that are determined during the calibration stage, at the beginning of the test.-We highlighted the EEG channels that provided the best detection performance indicators, in relation to the applied cognitive task.-We highlighted the synchronization of responses to certain EEG channels with the temporal profile of cognitive task application; this is a remarkable fact, especially in tests with an intermittent task.

Finally, in relation to the issue of finding the most relevant and reliable EEG channels that respond to the specific cognitive task, our research led to pertinent results from a neuroscientific point of view and in agreement with other studies reported in the literature. The results reported in the literature differ, but all mention nearby channels in the same regions of the cortex. Our results reinforce some findings in the literature and, in addition, thanks to the 64 electrodes used, the additional signals AF3, AF4, and AF8 in the frontal region, FT8 from the frontotemporal region, FC6 from the right central frontal, and TP7 and TP8 from the temporal–parietal were revealed. These results are quite consistent with various reports cited in the field [4,5,6,7,8,9,10,15]. Our study also has some limitations. The first is related to the limited number of experiments and types of cognitive load tests applied. The second is related to the offline application, in a signal preprocessing stage, of a systematic procedure for eliminating artifacts. The subsequent research effort will be oriented towards systematizing the experiments and ensuring uniformity of the testing conditions to discriminate as precisely as possible different types of cognitive tasks. On the other hand, we consider the potential of the developed detection algorithm for investigating other mental states using different narrowband frequencies.

## Figures and Tables

**Figure 1 sensors-25-03902-f001:**
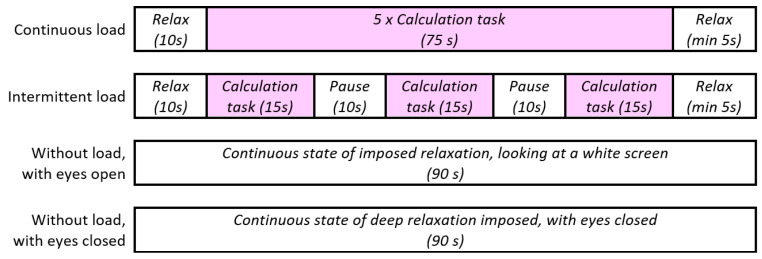
Test structure.

**Figure 2 sensors-25-03902-f002:**
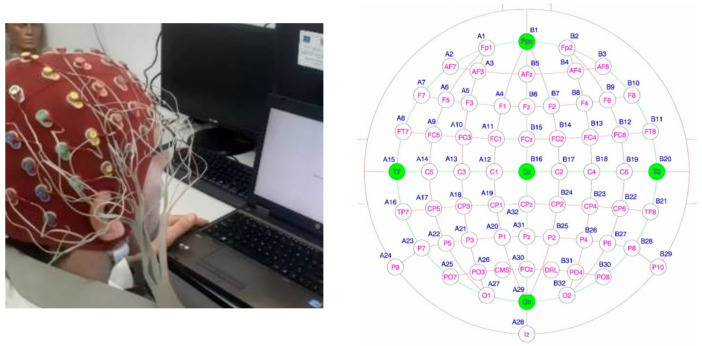
Experimental setup. The green points are used as references points to measure the brain activity.

**Figure 3 sensors-25-03902-f003:**
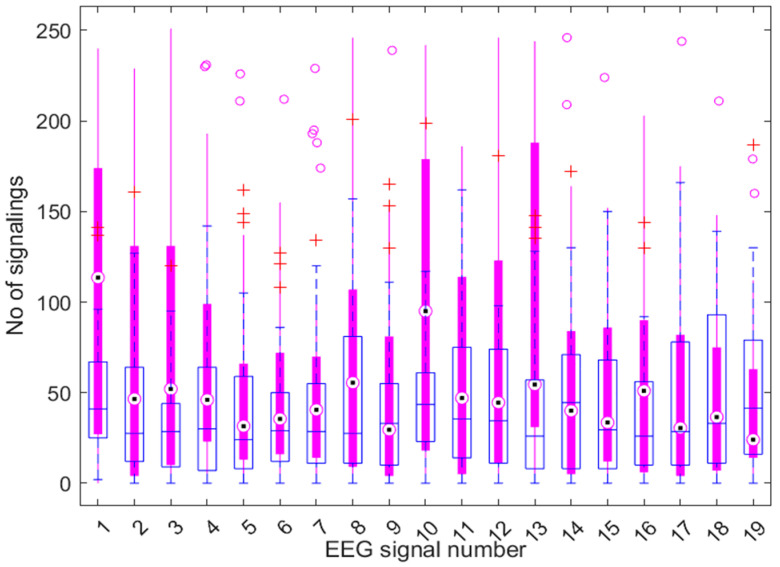
Statistical activity on EEG channels with box plot elements. Empty boxes and blue markers are for data without cognitive load, and filled boxes with red markers belong to data with mental computational load. Magenta circles mark outliers for cognitive load tests. Red + symbols mark outliers for tests without cognitive load. The thin lines in the extension of the boxes denote the range of values.

**Figure 4 sensors-25-03902-f004:**
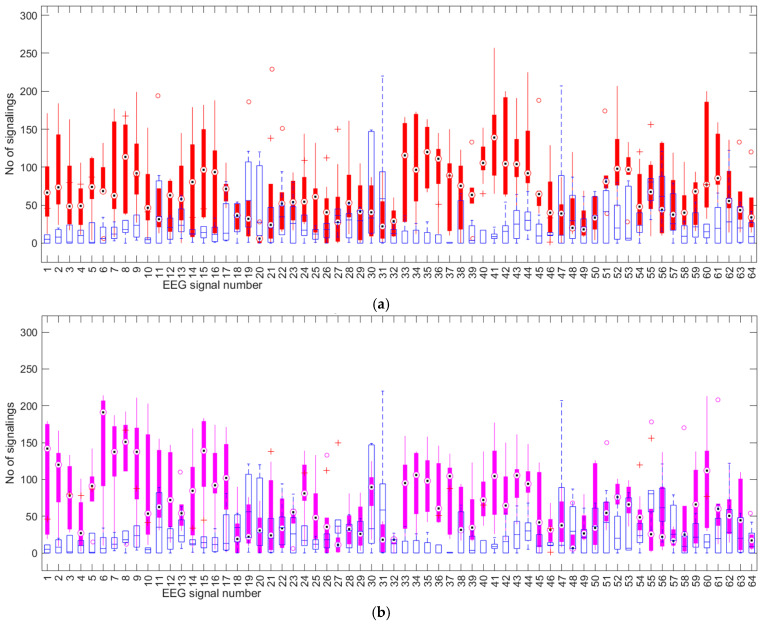
Statistical activity on 64 EEG channels: (**a**) for continuous load charge with red filled boxes and (**b**) for intermittent load charge with magenta filled boxes, both comparatively with the no-load state drawn with blue empty boxes. Magenta circles mark outliers for cognitive load tests. Red + symbols mark outliers for tests without cognitive load. The continuous and dashed thin lines in the extension of the boxes denote the range of values.

**Figure 5 sensors-25-03902-f005:**
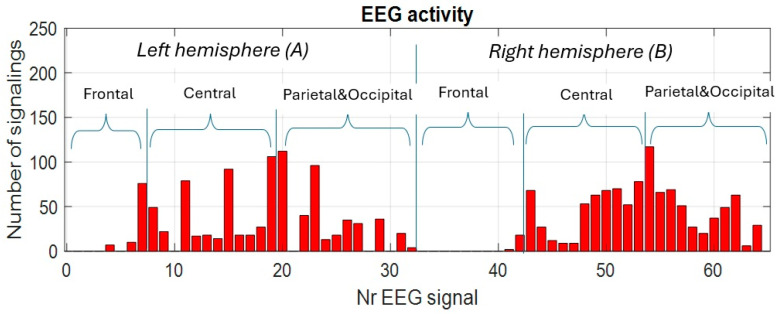
Activity on EEG channels for the no-cognitive-task test with eyes open (TW1).

**Figure 6 sensors-25-03902-f006:**
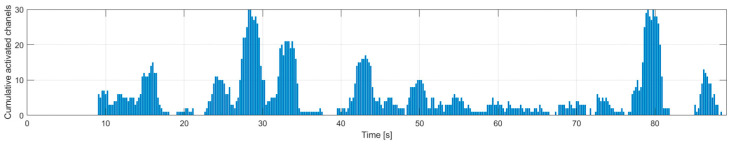
Cumulative activity on all EEG channels during the TW1 test. Denotes the fluctuation of the background state of the brain and due to other causes.

**Figure 7 sensors-25-03902-f007:**
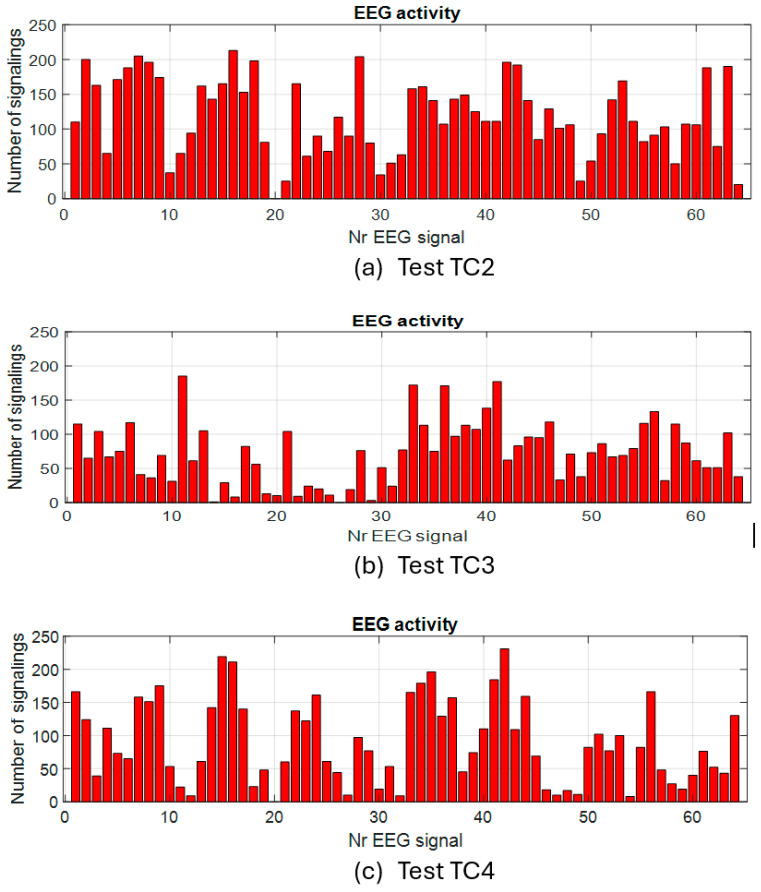
Activity on EEG channels for continuous cognitive task tests.

**Figure 8 sensors-25-03902-f008:**
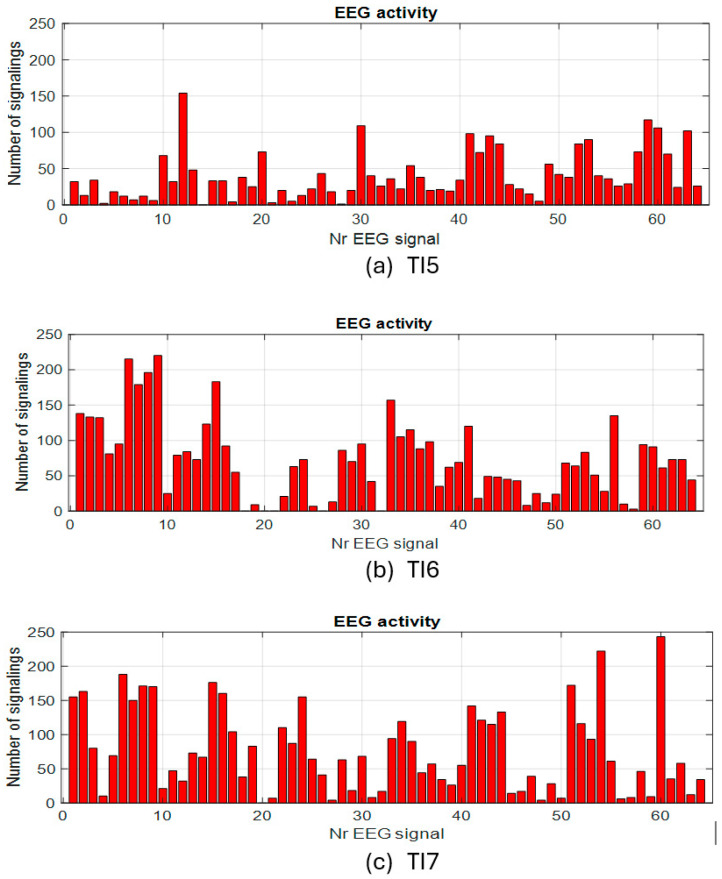
Activity on EEG channels for intermittent cognitive task tests.

**Figure 9 sensors-25-03902-f009:**
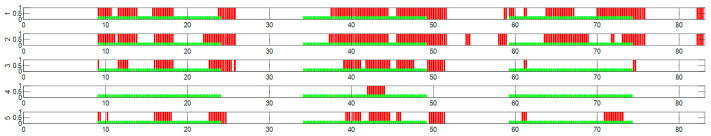
Response of EEG channels 1-5 in the presence of intermittent mental load captured from the analysis of the TI7 test. The red bars signify the detection of the load over time (represented in seconds on the abscissa), and the green lines indicate the intervals in which the cognitive load was applied.

**Figure 10 sensors-25-03902-f010:**
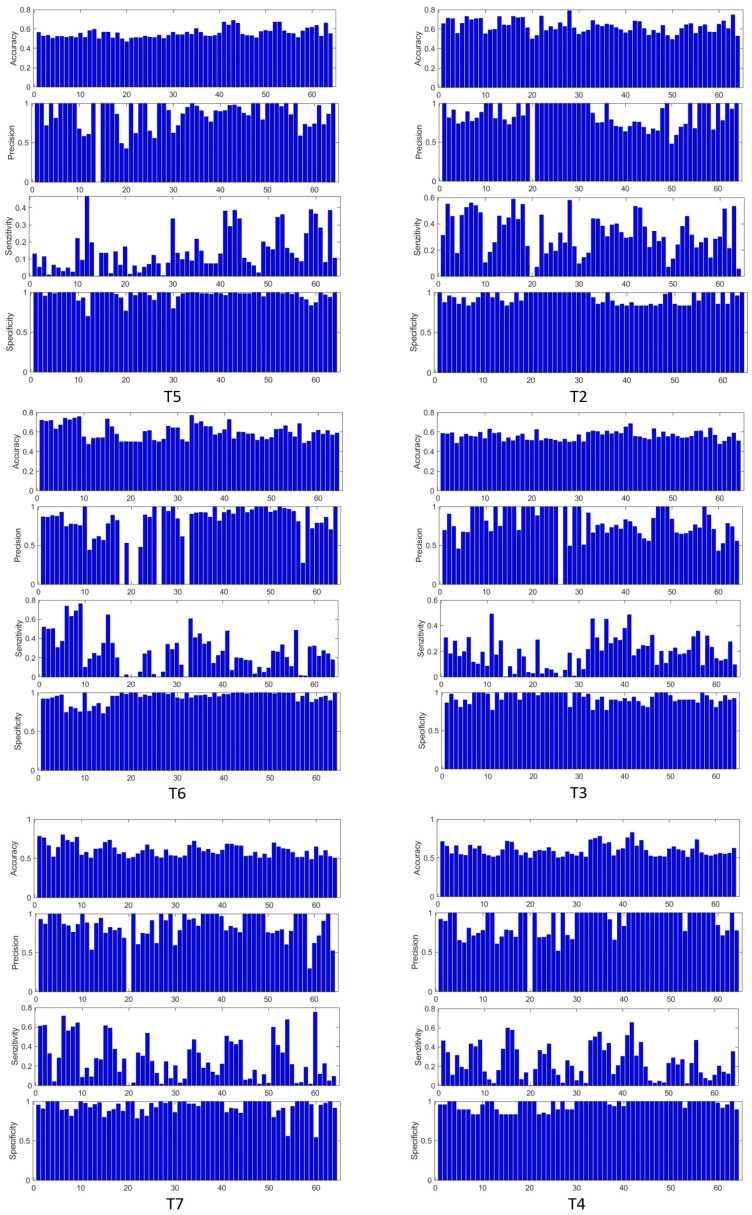
Accuracy, precision, sensitivity, and specificity indicators represented comparatively for all EEG channels in tests with intermittent load (left stack) and with continuous load (right stack).

**Figure 11 sensors-25-03902-f011:**
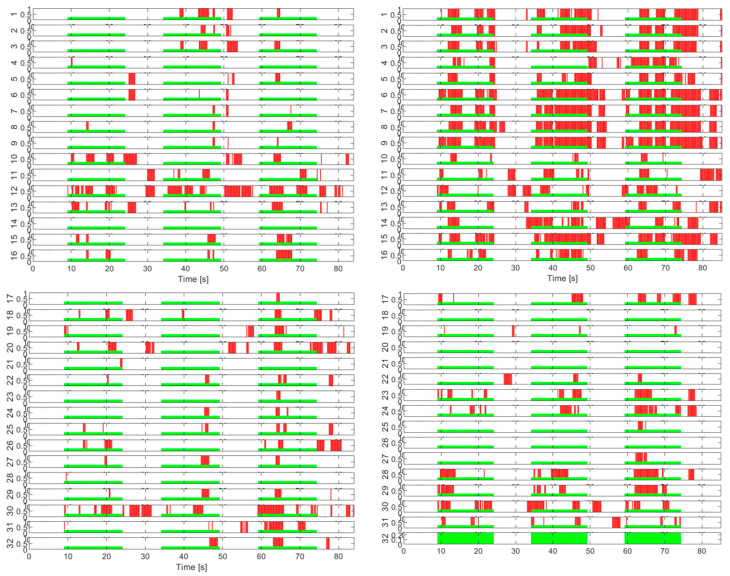
EEG channel activation patterns in T5 and T6 intermittent cognitive load tests. The light green areas indicate the periods of application of the cognitive load. The red areas indicate the detection of EEG activity in response to the load.

**Figure 12 sensors-25-03902-f012:**
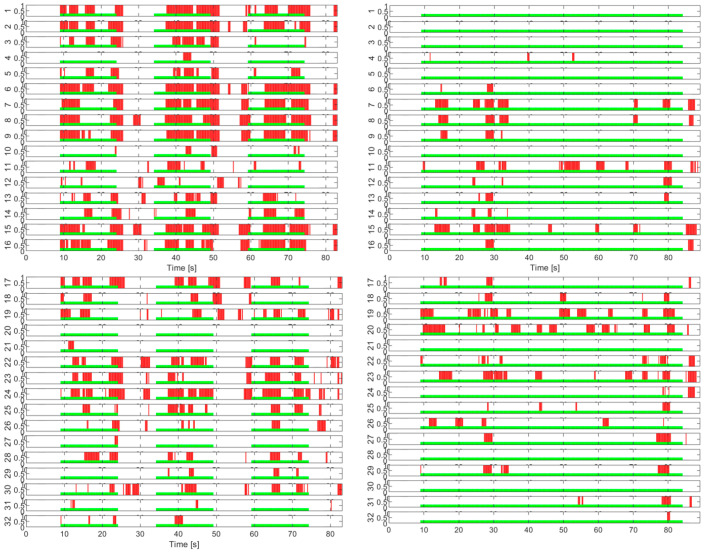
EEG channel activity presented in comparison between the T7 test—with intermittent cognitive task (left stack) and the reference test T1—relaxed state with eyes open (right stack).

**Figure 13 sensors-25-03902-f013:**
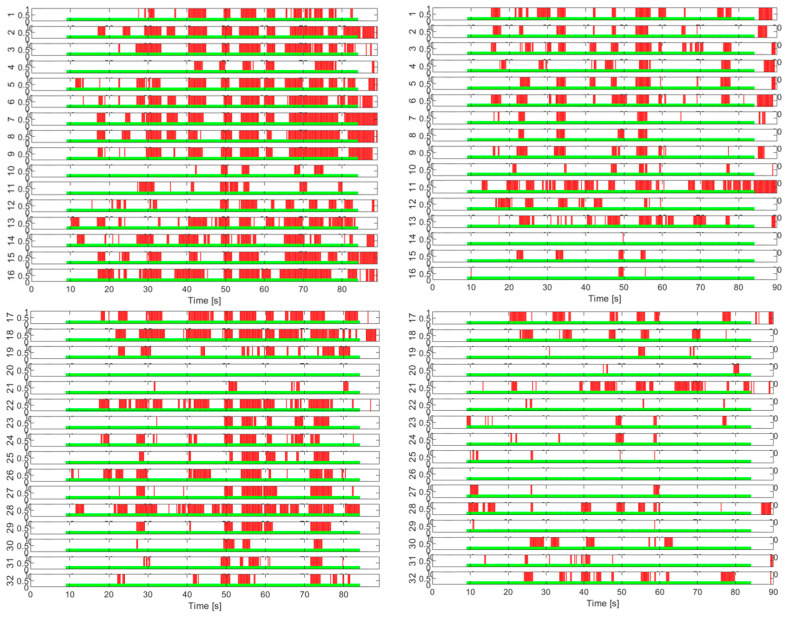
EEG activation patterns across channels in continuous cognitive load tests.

**Figure 14 sensors-25-03902-f014:**
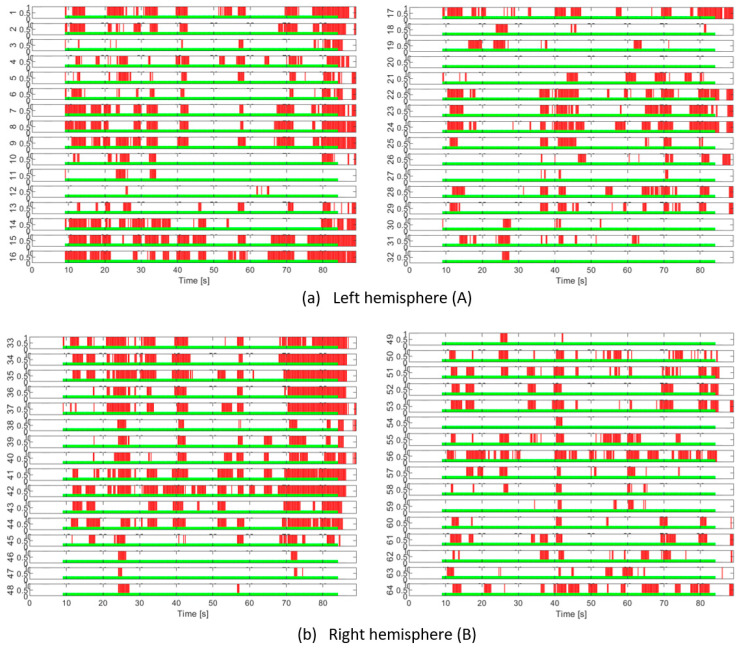
EEG activation patterns across channels for continuous cognitive load test T4.

**Table 1 sensors-25-03902-t001:** The significance of cognitive load detection.

EEG Channel		*t*-test Absolute Value	*p*-Value	Decision on Hypothesis H_1_ *	
No	Name			α = 0.05	α = 0.01
1	Fp1	4.133	0.0002	TRUE	TRUE
2	Fp2	2.228	0.0915	TRUE	FALSE
3	F3	4.045	0.0015	TRUE	TRUE
4	F4	2.577	0.0173	TRUE	FALSE
5	F7	1.010	0.4759	FALSE	FALSE
6	F8	1.593	0.1305	FALSE	FALSE
7	T3/T7	1.684	0.1428	FALSE	FALSE
8	T4/T8	0.949	0.3685	FALSE	FALSE
9	C3	0.699	0.5805	FALSE	FALSE
10	C4	3.282	0.0035	TRUE	TRUE
11	T5/P7	1.280	0.3209	FALSE	FALSE
12	T6/P8	1.972	0.0567	FALSE	FALSE
13	P3	3.453	0.0009	TRUE	TRUE
14	P4	0.785	0.4568	FALSE	FALSE
15	O1	0.627	0.6080	FALSE	FALSE
16	O2	1.603	0.0720	FALSE	FALSE
17	Fz	0.459	0.6355	FALSE	FALSE
18	Cz	0.000	0.9858	FALSE	FALSE
19	Pz	0.812	0.3954	FALSE	FALSE

* Critical values from t-table: 2.0317 for α = 0.05 and 2.7272 for α = 0.01.

**Table 2 sensors-25-03902-t002:** Significance of detection for continuous cognitive load versus no load.

EEG Channel		*t*-test Absolute Value	*p*-Value	Decision on Hypothesis H_1_ *		
No	Name			α = 0.05	α = 0.02	α = 0.01
1	Fp1	2.755	0.0469	TRUE	FALSE	FALSE
2	AF7	3.691	0.0210	TRUE	TRUE	FALSE
5	F3	3.794	0.0646	TRUE	TRUE	FALSE
6	F5	3.410	0.0129	TRUE	TRUE	FALSE
7	F7	3.015	0.0434	TRUE	FALSE	FALSE
8	FT7	3.170	0.1514	TRUE	FALSE	FALSE
9	FC5	3.285	0.0457	TRUE	FALSE	FALSE
10	FC3	2.880	0.0697	TRUE	FALSE	FALSE
12	C1	2.624	0.0675	TRUE	FALSE	FALSE
14	C5	2.875	0.0709	TRUE	FALSE	FALSE
15	T7	3.188	0.0224	TRUE	FALSE	FALSE
16	TP7	2.752	0.0435	TRUE	FALSE	FALSE
17	CP5	2.577	0.0569	TRUE	FALSE	FALSE
25	PO7	2.669	0.0612	TRUE	FALSE	FALSE
33	Fpz	3.636	0.0136	TRUE	TRUE	FALSE
34	Fp2	3.516	0.0161	TRUE	TRUE	FALSE
35	AF8	5.683	0.0011	TRUE	TRUE	TRUE
36	AF4	4.528	0.0053	TRUE	TRUE	TRUE
39	F2	2.912	0.0230	TRUE	FALSE	FALSE
40	F4	11.777	0.0004	TRUE	TRUE	TRUE
41	F6	5.073	0.0048	TRUE	TRUE	TRUE
42	F8	3.768	0.0091	TRUE	TRUE	FALSE
43	FT8	6.271	0.0044	TRUE	TRUE	TRUE
44	FC6	3.342	0.0145	TRUE	FALSE	FALSE
45	FC4	2.968	0.0315	TRUE	FALSE	FALSE
52	T8	3.7367	0.0126	TRUE	TRUE	FALSE
53	TP8	4.3212	0.0052	TRUE	TRUE	TRUE
58	P4	2.7750	0.0288	TRUE	FALSE	FALSE
60	P8	2.6361	0.0516	TRUE	FALSE	FALSE
61	P10	5.4085	0.0036	TRUE	TRUE	TRUE

* Critical values from t-table: 2.5706 for α = 0.05, 3.3650 for α = 0.02, and 4.0321 for α = 0.01.

**Table 3 sensors-25-03902-t003:** Significance of detection for intermittent cognitive load versus no load.

EEG Channel		*t*-test Absolute Value	*p*-Value	Decision on Hypothesis H_1_ *		
No	Name			α = 0.05	α = 0.02	α = 0.01
1	Fp1	3.293	0.0184	TRUE	FALSE	FALSE
2	AF7	4.735	0.0092	TRUE	TRUE	TRUE
5	F3	3.000	0.0289	TRUE	FALSE	FALSE
6	F5	4.396	0.0081	TRUE	TRUE	TRUE
7	F7	4.421	0.0075	TRUE	TRUE	TRUE
11	FC1	6.136	0.0986	TRUE	TRUE	TRUE
12	C1	2.650	0.0375	TRUE	FALSE	FALSE
15	T7	3.962	0.0100	TRUE	TRUE	FALSE
16	TP7	3.399	0.0143	TRUE	TRUE	FALSE
25	PO7	2.705	0.0661	TRUE	FALSE	FALSE
33	Fpz	3.601	0.0139	TRUE	TRUE	FALSE
34	Fp2	4.350	0.0072	TRUE	TRUE	TRUE
35	AF8	5.291	0.0035	TRUE	TRUE	TRUE
36	AF4	5.098	0.0151	TRUE	TRUE	TRUE
39	F2	2.645	0.0774	TRUE	FALSE	FALSE
40	F4	9.747	0.0120	TRUE	TRUE	TRUE
41	F6	4.694	0.0108	TRUE	TRUE	TRUE
42	F8	2.717	0.0284	TRUE	FALSE	FALSE
43	FT8	5.204	0.0034	TRUE	TRUE	TRUE
44	FC6	5.058	0.0044	TRUE	TRUE	TRUE
45	FC4	2.661	0.1205	TRUE	FALSE	FALSE
52	T8	3.691	0.0156	TRUE	TRUE	FALSE
61	P10	2.608	0.1044	TRUE	FALSE	FALSE

* Critical values from t-table: 2.5706 for α = 0.05, 3.3650 for α = 0.02, and 4.0321 for α = 0.01.

**Table 4 sensors-25-03902-t004:** EEG channels that become significant with decreasing confidence level.

Level of Confidence	Cognitive Load		
	Continuous	Intermittent	Physionet
95%	Fp1, F7, FT7, FC5, FC3, C1, C5, T7, TP7, CP5, PO7, F2, FC6, FC4, P4, P8	Fp1, F3, C1, PO7, F8, FC4, P10	Fp1, Fp2, F3, F4, C4, P3
98%	AF7, F3, F5, Fpz, Fp2, F8, T8	T7, TP7, Fpz, T8	
99%	AF8, AF4, F4, F6, FT8, TP8, P10	AF7, F5, F7, Fp2, AF8, AF4, F4, F6, FT8, FC6	Fp1, F3, C4, P3

**Table 5 sensors-25-03902-t005:** Values of performance indicators for tests with intermittent cognitive load.

EEG Channel		Precision	Specificity	Sensitivity	Accuracy
2	AF7	0.943–0.815	0.980–0.875	Max 0.550	0.713–0.581
6	F5	1–0.716	1–0.958	Max 0.500	0.707–0.555
7	F7	0.976–0.649	0.903–0.854	Max 0.500	0.660–0.544
34	Fp2	1–0.750	1–0.854	Max 0.500	0.720–0.645
35	AF8	1–0.754	1–0.875	Max 0.500	0.700–0.629
36	AF4	1–0.700	1–0.840	Max 0.500	0.685–0.652
40	F4	0.880–0.637	0.969–0.833	Max 0.500	0.750–0.563
41	F6	1–0.696	1–0.854	Max 0.500	0.760–0.594
43	FT8	0.890–0.637	0.833–0.800	Max 0.500	0.770–0.563
44	FC6	0.860–0.800	0.865–0.820	Max 0.600	0.670–0.530

## Data Availability

The data presented in this study are available on request from the corresponding author, due to the restriction of their use for commercial or non-academic purposes.

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
