# Peer review of "Narrowband Theta Investigations for Detecting Cognitive Mental Load"

_sensors, 2025, doi:10.3390/s25133902_

Round 1

Reviewer 1 Report

Comments and Suggestions for Authors

This study found a new EEG signal metric for detecting cognitive load in the theta band, but there are some major concerns that make it hard to accept in its current form. Most critically, the entire analysis is based on just one participant, with no replication or statistical validation, which seriously limits any generalizability. On top of that, the objective is vague and hard to follow, and the preprocessing pipeline is under-described. There’s not enough detail, for example, on filtering (like filter type, order, window type, and so on), artifact rejection, or whether baseline correction was even applied, or from what steps FieldTrip was used, and how the other steps were implemented. There is no real discussion. While the title is named results and discussion, I don’t find any in-depth discussion. It simply includes reports. Overall, the idea is interesting, but without description of essential details on preprocessing and data from more than one person, it’s hard to draw any meaningful conclusions.

Author Response

Some (partial) improvements were in revised manuscript attached. 

Definitely, your observations are right. 

I will try to obtain a new term from editor.

Thank you!

Reviewer 2 Report

Comments and Suggestions for Authors

I would like to thank the authors for this quality of reseach. The paper reads well and brings something thing new to the fields of medical signal processing. It's well written and the authors focused on the main work. The study focuses on using EEG signals, specifically in the theta band (4-7 Hz), to detect cognitive mental load induced by arithmetic tasks.

  • The authors  introduced a new metric called the length of the signal curve (Lsc) and compared it with energy metrics.  The introduction of the Lsc as a signal metric for narrow-band theta analysis is innovative and well-motivated. Its independence from signal bias. But the single-subject design and lack of statistical power are major drawbacks.
  • The identification of specific EEG channels (e.g., Fp1, AF7, AF3) as reliable for cognitive load detection aligns with neuroscientific literature and provides actionable insights for electrode optimization.
  • The artifact handling method and the choice of calibration period could also be potential areas for critique.
  • Were alternative artifact removal techniques (e.g., ICA, DWT denoising) considered? How was the threshold for spike detection determined?

  • The calibration period (tcalib=10s) is brief. Was the stability of the baseline state during calibration verified? How sensitive is the algorithm to transient fluctuations in the reference state?
  • The synchronization of EEG responses with task timing (e.g., Figure 7) is compelling.

Thank you.

Comments on the Quality of English Language

The English could be improved

Author Response

Thank you for your revision and useful suggestion. My answers are punctual in following:

  • But the single-subject design and lack of statistical power are major drawbacks.

Here we made a statistical approach with records from open access Physionet database, and also I have added more tests in our experiment. (see the submitted revised manuscript)

  • The artifact handling method and the choice of calibration period could also be potential areas for critique.
  • Were alternative artifact removal techniques (e.g., ICA, DWT denoising) considered? How was the threshold for spike detection determined?

Our goal was to develop and test a cognitive task detection algorithm in as much real-time as possible. At this stage, we excluded ICA and other techniques that are usually used off-line (or are time-consuming even in fast versions), preferring a simpler algorithm (somewhat wavelet-based). In principle: atypical signal peaks are eliminated (with replacement) with the proposed model (based on equations 10 and 11), the slow component (below 2Hz) is extracted by subtraction from the raw signal, and (very) narrow-band filtering avoids some artifacts. All of this contributes to the reduction of biological and nonbiological artifacts.

  • The calibration period (tcalib=10stcalib​=10s) is brief. Was the stability of the baseline state during calibration verified? How sensitive is the algorithm to transient fluctuations in the reference state?

This question is just! For some channels, in some tests this could hapend . The time tcalib is an adjustable parameter...it could be set longer. For instance , with the longer records from Physionet I used 30s to 50s . Slight improvement was observed.

  • The synchronization of EEG responses with task timing (e.g., Figure 7) is compelling.
  •  
  • Yes, indeed, and this was confirmed in suplementary tests thar we added. 

Reviewer 3 Report

Comments and Suggestions for Authors

Recommendation: Minor

Comments: Frequency domain selection for EEG signal analysis relies on neuroscience evidence and practical signal processing considerations. The acquisition method and intensity variations of EEG signals are essential for advancing neuroscience research and the development of brain-computer interface technology. In this article (sensors-3584130), the author sought to develop an EEG signal analysis algorithm capable of detecting specific cognitive tasks in real time. The approach avoids machine learning techniques that rely on extensive training data, instead focusing on identifying the most reliable electrical brain channels associated with cognitive tasks. Here are some suggestions for this article.

  1. It is advisable that the written expression in the paper should steer clear of colloquial language.
  2. The authors should highlight the advantages of reported method, particularly in comparison to machine learning, which is widely regarded as a promising technology.
  3. In the Methodology section, comprehensive details about the EEG testing equipment and the experimental procedures should be provided to ensure readers to understand.
  4. Please make the necessary modifications to Figure 1.
Comments on the Quality of English Language

No

Author Response

Thank you for your revision and for your useful comments!

I am going to do the recommended improvements for the final form of the paper. 

At this moment I submit to you the partial revised manuscript.

Round 2

Reviewer 1 Report

Comments and Suggestions for Authors

As the authors acknowledged, the revised manuscript includes only partial improvements and does not yet address all the issues raised.

Author Response

On top of that, the objective is vague and hard to follow, and the preprocessing pipeline is under-described. There’s not enough detail, for example, on filtering (like filter type, order, window type, and so on), artifact rejection, or whether baseline correction was even applied, or from what steps FieldTrip was used, and how the other steps were implemented

Response (green highlighted in text): 

In the data preprocessing stage, steps (3)-(5) are effectively considered. Thus, in the off-line working mode, in step (3), the data were read using the ft_read_header and ft_read_data function pair from the FieldTrip toolbox, after which they were normalized using the z-score method, in the range [-1, 1]. Then, in step (4) the low-pass filter was effectively applied in the baseline correction and also to exclude certain biological and non-biological artifacts. The low-pass filter is of the digital IIR Butterworth type with a pass frequency of 2Hz and a cutoff frequency of 2.25Hz. It extracts the slow signal which is then subtracted from the raw signal. In step (5) narrow-band filtering specific for mental load detection is applied. A high-order FIR digital filter, more precisely 4000, with a narrow passband from 4 to 5 Hz was used. This results in the signal to be analyzed for mental load detection following steps (6)-(12). At step (13) the calibration is already completed, and the algorithm continues its execution cyclically, on successive signal windows with the application of the atypical peak artifact elimination model, according to equations (10) and (11) and the testing of the load conditions with equations (8) and (9). The answer consists of the number of occurrences of the detected cognitive load, which are counted and displayed graphically for all EEG channels.

There is no real discussion. 

Response (green highlighted in text):

These results are encouraging for our stated goal in the introduction of the paper, i.e. to detect cognitive load associated with arithmetic calculation without involving machine learning techniques and large data sets. The presented experiments proved that the detection of mental load based on an individual threshold - as a reference state, using a distinct signal metric, and analyzing the signals in a specific narrow band, is feasible. The discussion of the results involves several panels.

First, the operation of the method was proven on the Physionet database and then on the data sets acquired through our own experiments. In both cases, the method proved capable of discriminating mental load in relation to the reference resting state despite the differences in equipment and test format. Regarding the performance of the load detector, we found that the duration of the calibration period positively influences the detection accuracy, but a too long duration may include disturbances in the individual's resting state, which was found in some recordings from Physionet where the last 40 seconds of rest before applying the mental load were taken for calibration. In our experiments, the subject undergoing tests was better controlled, in the sense of respecting the state imposed by the experiment protocol. This being in this case an advantage of performing the tests on a single individual. The duration of the reference state imposed for calibration was only 10 seconds. Another parameter that influences the accuracy of the method is the size of the signal window on which the metric used for detection is averaged. The window comprises a specified number of signal periods over which the averaging is done, and the size of this number is the subject of a compromise: a short window benefits real-time detection but degrades detection accuracy, and, conversely, a longer window increases accuracy but introduces a corresponding delay. We obtained the presented results using a signal frame containing 10 periods, which means about 2.2 seconds of response delay and the best detection accuracy around 0.7 (as seen in Table 5). Increasing the frame length to 20 periods results in an inherent delay of over 4 seconds, but also an increase in accuracy to about 0.85.

Secondly, the relevance of the results from an experimental point of view is supported by the evident synchronization of the activation of some signals with the profile of the application of the cognitive task over time. Thus, our experiments were structured in such a way as to stimulate specific mental activity dynamically and intermittently in the narrow band chosen for analysis.

Third, the sustainability of the results from a neuroscience perspective comes from the statistical analysis of the experimental data which revealed a high level of confidence for certain signals with a confirmed role in the detection of mental load, as presented in Table 4.

Reviewer 2 Report

Comments and Suggestions for Authors

All the requested revisions are included. Thank you 

Author Response

I have added additional clarifications to the text (highlighted in green).
